# Spatial Interaction Spillover Effects between Digital Financial Technology and Urban Ecological Efficiency in China: An Empirical Study Based on Spatial Simultaneous Equations

**DOI:** 10.3390/ijerph18168535

**Published:** 2021-08-12

**Authors:** Yaya Su, Zhenghui Li, Cunyi Yang

**Affiliations:** 1School of Finance, Hunan University of Technology and Business, Changsha 410205, China; suyaya@hnu.edu.cn; 2Guangzhou Institute of International Finance, Guangzhou University, Guangzhou 510405, China; 3School of Economics and Statistics, Guangzhou University, Guangzhou 510006, China

**Keywords:** digital financial technology, urban ecological efficiency, spatial simultaneous equations, GS3SLS, interaction, spillover

## Abstract

As a core component of the digital economy, digital financial technology has a complex interactive and interdependent relationship with ecological efficiency. From the holistic spatial interaction perspective, this paper uses spatial simultaneous equations and generalized spatial three-stage least squares (GS3SLS) to analyze the spatial interaction spillovers between digital financial technology and urban ecological efficiency based on data from 284 Cities in China from 2008 to 2018. The results show that: (1) Digital financial technology and urban ecological efficiency promote each other, and the latter is relatively dominant. (2) Both digital financial technology and urban ecological efficiency have significant spatial spillover effects. (3) Digital financial technology in surrounding cities has a restraining effect on local ecological efficiency, and the improvement of ecological efficiency in surrounding cities has a siphon effect on local digital financial technology. (4) There is spatial and period heterogeneity in the intensity of the spatial interaction spillover effect between the two. With resources and environment increasingly becoming rigid constraints on economic growth, these findings help identify new drivers of regional ecological efficiency and promote the coordinated development of digital finance and green ecology.

## 1. Introduction

With the advent of the digital era, Internet enterprises, technology companies and financial technology institutions actively take advantage of digital technology to empower finance. They constantly create new business models, promoting the transformation and upgrading of traditional financial institutions and enhancing the driving ability of digital financial technology to economic development. At the same time, the resource and environmental constraints caused by rapid economic development are becoming increasingly prominent, and the environmental problems are attracting more and more widespread attention. In this context, based on seeking the coordination and unity of economic development, resource conservation and environmental protection, optimal ecological efficiency has become the only way to achieve high-quality economic development, i.e., to realize the highest economic output with minimum resource consumption and environmental impact [1,2]. Schaltegger and Sturm first proposed ecological efficiency, namely the ratio of the added value to the environmental impact, and it has been used extensively since then [3]. Ecological efficiency reflects the level of green development, emphasizes the unified and coordinated development of economic development and environmental protection, and can measure the coordination degree of the environment, resources and economic development [4,5]. Therefore, it is of profound significance to study the influencing factors of ecological efficiency and its spatial distribution.

Green development cannot be separated from the support of financial resources. As the combination of the financial industry and digital technology, like big data, cloud computing, artificial intelligence, 5G technology, and blockchain technology, digital financial technology plays a crucial role in optimizing ecological efficiency [6]. Studies on the relationship between digital financial technology and ecological efficiency have found that there may be complex interactions between the two. On the one hand, the impact of digital financial technology on ecological efficiency is uncertain. Digital financial technology helps to fully play the resource allocation effect and innovation effect, which can enhance the traditional financial industry’s ability to serve the real economy and boost the asset management business from the virtual to the real, creating objective and realistic conditions for optimizing ecological efficiency. However, the development of digital financial technology brings rapid economic growth while consuming more resources and increasing pollution emissions. On the other hand, the improvement of ecological efficiency also affects the development of digital financial technology. At present, optimizing ecological efficiency has become the consensus goal of various governments, playing a particular guiding role in the flow of funds. With the trend of green production, the effects of resource aggregation and business innovation are becoming more and more prominent. The industry is transforming from energy-intensive to knowledge-technology-intensive, and digital financial technology has made significant progress. Existing research focuses more on the one-way relationship between digital financial technology and ecological efficiency. With the rapid development of digital financial technology and the improvement of green development demands, it is necessary to test the interaction between the two from a holistic perspective.

From the perspective of spatial dimension, the differences of regional development foundation and environment, together with the guidance of the development mode “From points to areas; parts pushing the whole”, make spatial heterogeneity the characteristic fact of digital financial technology and green development, which is manifested as the spatial agglomeration of digital financial technology and ecological efficiency [7,8]. However, the spatial interaction spillover effect between the two has not been paid much attention by the researchers. The new economic geography theory and the spatial measurement model fit the reality of the cross-regional role of the influencing factors in the open state, providing powerful technical support for exploring the complex relationship between digital financial technology and regional ecological efficiency. As promoting the development of digital financial technology and improving ecological efficiency become more and more prominent, in-depth exploration of the relationship between digital financial technology and regional ecological efficiency in a state of spatial interaction based on a holistic perspective has become an increasingly important scientific issue. Can the development of digital financial technology under the state of spatial interaction improve regional ecological efficiency? Does ecological efficiency boost the progress of digital financial technology? How to realize the coordinated development of digital financial technology and regional green ecology? This paper will sort out and analyze the above problems.

Existing literature suggests that digital financial technology and ecological efficiency interact in complex ways. Previous studies on the impact of digital financial technology on ecological efficiency mainly focus on the following aspects: innovation of financial instruments, related research focuses on green funds and green bonds promoting ecological efficiency [9,10,11,12,13,14,15]; in terms of environmental risk management, environmental pollution liability insurance is found to be an appropriate measure [16,17]; for financial institutions, they establish new financial markets, reduce risks faced by enterprises and social environment and improve ecological efficiency [18,19,20,21]; digital financial technology can improve marine ecological benefits and is becoming more evident in the southern coastal provinces of China [6]. In addition, Huang et al. and Flaherty et al. also analyze the mutual promotion relationship between financial development and regional green development, financial aggregation, and ecological efficiency from different perspectives [1,2]. Previous studies on the impact of ecological efficiency on finance are relatively rare. It is mainly reflected in the impact of ecological efficiency on economic development. With the development of agricultural ecology, ecological and economic benefits have been significantly improved, and social benefits have also been improved accordingly [22]. Increasing ecological efficiency and reducing direct energy loss can save energy and reduce the investment possibilities of primary energy consumption in fuel, thus improving economic benefits [23,24]. The Reduction Principle aims to minimize primary energy, raw materials, and waste inputs by improving ecological efficiency and consumption processes (introducing better technology or more compact, lighter products, simplified packaging, more efficient household appliances) to realize a simpler lifestyle [25,26]. In terms of production, Figge et al. identify two primary ways for enterprises to improve ecological efficiency in the production process: maintaining or increasing the product’s value and reducing the impact on the environment. This can be achieved by using fewer resources per unit output and replacing hazardous substances [27]. Ecological efficiency is mainly a commercial concept that focuses on the economic and environmental aspects of sustainability while neglecting the social aspects [28,29,30,31].

The development of digital financial technology has spatial characteristics. Knight and Wójcik describe financial technology as a controversial, mature and geographical research field. By the end of June 2020, 10 financial technology papers (classified by the Web of Science) had been published in the field of geographic research [7]. Clark and O’Connor oppose the assertion of “the end of geography” and use information-related factors to validate the complex division of financial labor at different spatial scales [32]. Shen et al. construct a comprehensive index of digital financial inclusion, and the spatial distribution of the index presents a strong regional aggregation and the aggregation pattern of national income groups [33]. To obtain the economic intensity of agglomeration similar to that of the world’s top cities in the financial field, Shanghai’s spatial structure needs to be more concentrated, and its economic density needs to be more complex [34]. Rodima and Grimes analyze the latest developments of international remittances in developing countries from the perspective of infrastructure, which reveals essential junction points between diverse money transfer pathways and institutions, depicting their spatial configuration and relationality as well as their potential to affect power differentials, and allowing for a socially embedded view of digital disruption [35].

The development of ecological efficiency also has spatial characteristics. A large number of studies are based on the spatial effects of ecological efficiency. The study of Xu et al. shows that ecological benefit spillover networks in various regions exhibit a typical core-edge structure, and there is a distinct hierarchical structure among the blocks with different directions and functions [8]. Ecological efficiency has a spatial auto-correlation, and there is a spatial aggregation effect between provincial financial development and ecological efficiency [36]. Chen et al. found that industrial agglomeration, pollution and ecological efficiency have significant spatial spillover effects, and agglomeration has a significant inverted U relationship with wastewater discharge, sulfur dioxide emission and soot discharge, and a significant U type relationship with ecological efficiency [37]. China’s regional ecological efficiency fluctuates, but the overall improvement is noticeable, and the gap between regions is further widened [38]. Ren et al. believe that China’s overall ecological efficiency is still at a low level, with significant differences between different regions; the eastern ecological efficiency is the highest, followed by the central region, and the gap between the central and western regions is gradually narrowing [39].

Based on the literature review, we can find many deficiencies in the previous research. First, previous research mainly focuses on the impact of digital financial technology on ecological efficiency, easy to overlook its endogeneity; few scholars study the relationship between them. Second, both the development of digital financial technology and the development of ecological efficiency have prominent spatial characteristics. The empirical evidence of previous studies mostly ignores the spatial interaction effect between the two, and the result analysis is inevitably biased. Finally, the number of samples studied in previous research is usually small, and the time span is insufficient, so it is difficult to fully reflect the interaction mechanism between the two during the long-term development process. Therefore, it is necessary to carefully study the spatial interaction spillover effect between digital financial technology and ecological efficiency.

## 2. Theoretical Framework

At present, China is in a stage of rapid development in digital financial technology, and the resource consumption and pollution caused by the rapid growth of the industrial economy are also criticized by other countries [40,41]. At the same time, China is also the net recipient of foreign spillovers most of the time [42,43,44]. As China’s economic structural transformation and economic development enter a new normal, digital financial technology is also urgently needed to provide new momentum for a new round of industrial revolution. The development relationship between China’s digital financial technology and ecological efficiency is of crucial representative significance. Based on the critical scientific issues raised above, this paper uses the data of more than 200 Chinese cities for 11 years and test the spatial interaction spillover effect between digital financial technology and urban ecological efficiency with the help of spatial simultaneous equations and GS3SLS.

Figure 1 shows the logical structure of this study, specifically as follows:

In Section 1, through reviewing existing relevant literature, the core issue of the research is proposed: what is the relationship between digital financial technology and urban ecological efficiency? In Section 2, based on the analysis of the relationship between key variables, this paper puts forward the research hypotheses and introduces the empirical model, data sources, and pre-processing. Section 3 analyzes the spatial interaction spillover effect between digital financial technology and urban ecological efficiency and tests its robustness. Section 4 is an additional analysis, which considers the heterogeneity of spatial interaction spillover effects based on regional differences and time differences. Section 5 draws the basic conclusions.

The main work and marginal contribution of this paper are as follows. Based on the public data of 284 prefecture-level cities in China from 2008 to 2018, this paper calculates the digital financial technology index and urban ecological efficiency using the Python network crawler and Super-SBM-GML model. Through spatial simultaneous equations and the GS3SLS estimation method, this paper analyzes the spatial interaction spillover effect and its heterogeneity between digital financial technology and urban ecological efficiency. The empirical results show that: (1) there are mutual promotion effects between digital financial technology and urban ecological efficiency, and the latter is in a relatively dominant position. The promotion of ecological efficiency by digital financial technology is in line with the previous research [9,10,11,12]. The research in this paper makes up for the blank in academic circles on the impact of ecological efficiency on digital financial technology. In other words, based on the previous research about the impact of ecological efficiency on economic development, this paper expands the impact object of ecological efficiency [22,23,24]. (2) Both digital financial technology and urban ecological efficiency have significant spatial spillover effects; that is, digital financial technology and urban ecological efficiency of surrounding cities promote local digital financial technology and urban ecological efficiency, respectively. This is not only consistent with the research results of Shen et al. on the spatial distribution of digital financial indices [33], but also consistent with the research results of Xu et al. on the ecological efficiency spatial spillover [8]. (3) Digital financial technology of surrounding cities has an inhibitory effect on the ecological efficiency of local cities, and the improvement of the ecological efficiency of surrounding cities has a siphon effect on the local digital financial technology. Most previous studies have ignored the spatial interaction effect between the two, and the research results of this article make up for the vacancy here. (4) There is temporal and spatial heterogeneity in the intensity of the spatial interactive spillover effects between digital financial technology and urban eco-efficiency; that is, the intensity of the spatial interactive spillover effects between the two are different in the eastern, central, and western regions as well as in different periods. The heterogeneity results in this paper are similar to the research of Liu et al. and Shi et al. [45,46], and the results are more comprehensive compared with other research, where they only considered one-way spatial relationship between variables.

## 3. Research Design and Pretreatment

### 3.1. Theoretical Analysis and Research Hypotheses

Digital financial technology can help improve spatial allocation and utilization efficiency of innovative resources, further improving urban ecological efficiency. The improvement of urban ecological efficiency can also promote financial innovation, absorb more financial resources, and improve digital financial technology through paths like industrial ecosystem optimization. In other words, there is an internal mechanism of mutual promotion between the two, which needs to be revealed through systematic research. Firstly, digital financial technology can optimize the momentum of urban green development and improve urban ecological efficiency through innovation, sale economy, knowledge spillover and environmental effect. Secondly, the improvement of urban ecological efficiency means that the resources invested in urban development produce more benefits than before. The improvement of urban ecological efficiency can promote the spatial aggregation of digital financial technology resources through resource aggregation, business form innovation, cost reduction and environmental optimization. Based on the above analysis, this paper proposes the following hypothesis:

**Hypothesis 1** **(H1).**
*There are mutual promoting effects between digital financial technology and urban ecological efficiency.*


As the scale of resource flows across regions increases and the interregional competition for innovation resources intensifies, the digital financial technology and urban ecological efficiency of specific regions are inevitably affected by neighboring regions. From a digital financial technology perspective, firstly, financial technology companies (or financial industry companies with financial technology departments) are the principal inventors and primary sources of supply of advanced digital financial technology. Financial enterprises realize their technology transfer through cross-regional direct investment internalization. This act of technology transfer brings an external economy to the target city, i.e., the spillover of digital financial technology. Meanwhile, due to reasons like customer stickiness, the expansion of financial enterprises often follows the principle of proximity; that is, they first deploy in the cities around the headquarters, then gradually expand the radiation area, spatially presenting a distribution of aggregation towards the headquarters. Secondly, improving digital financial technology in specific regions helps benefit the surrounding areas through talent and knowledge flow, innovation and demonstration effect, forming a spatial development thrust from the center to the periphery and driving the development of digital financial technology in neighboring areas. Thirdly, the industrial policies formulated by local governments often cover a province or a city cluster; digital financial technology as one of the main targets of policy support in recent years reflects the differences between regions, and the development level of digital financial technology is therefore spatially related.

From the perspective of urban ecological efficiency, firstly, the improvement of ecological efficiency of neighboring cities can improve the ecological efficiency of local cities through innovation and demonstration effect and knowledge spillover effect. Secondly, urban ecological efficiency is naturally related to the urban environment. The natural environment of neighboring cities is similar to local cities, and other objective conditions also have a high spatial correlation, leading to a certain degree of similarity in the development trend of ecological efficiency. Based on the above analysis, this paper puts forward the following hypothesis:

**Hypothesis 2** **(H2).**
*Both digital financial technology and urban ecological efficiency have significant spatial spillover effects.*


The spatial interaction mechanism of digital financial technology and urban ecological efficiency is complex. First, as an essential driver of regional innovation and development, the agglomeration of digital financial technology to the surrounding cities weakens the local development momentum. Interregional digital financial technology competition exerts downward pressure on local ecological efficiency. Areas with lagging financial development need to actively undertake resource transfer, strengthen the endogenous support mechanism of local financial development for urban ecological efficiency, and improve the systematic driving force of local urban ecological efficiency. Second, the optimization of the industrial development environment and the innovative development of the ecosystem driven by the improvement of ecological efficiency of surrounding cities can siphon local financial resources. At the same time, surrounding cities with high ecological efficiency often have more sound and more substantial financial support policies, forming a powerful attraction to the local digital financial technology enterprises and talents. Based on the above analysis, the following hypothesis is proposed:

**Hypothesis 3** **(H3).**
*The digital financial technology of surrounding cities has an inhibitory effect on the ecological efficiency of local cities, and the improvement of ecological efficiency of surrounding cities has a siphon effect on local digital financial technology.*


Figure 2 comprehensively shows the hypotheses H1, H2 and H3 in this paper.

Digital financial technology and urban ecological efficiency involve factors such as international and domestic environment, geographical location, policy strategy and resource distribution in each region, so the influencing ways and effects between digital financial technology and urban ecological efficiency are different. On the one hand, according to the economic and social development levels of provincial administrative regions, China can be divided into three economic zones, and all cities are divided into the eastern, central and western regions, accordingly. It is generally believed that different regions have different advantages and development modes. The eastern region has the coastal location advantages to develop tertiary industry and intensive agriculture and develop products in high quality, precision, and cutting-edge. The central region has the advantages of building national energy and raw material base, building agricultural production, circulation and processing base, and strengthening transportation construction and ecological environment construction. The western region should improve the agricultural ecological environment, stabilize farmland area, increase yield per unit area, develop energy and minerals and become the power base in China. Thus, it can be seen that digital financial technology and ecological efficiency must be heterogeneous among cities in the three economic zones, and the intensity of their spatial interaction spillover effects may also be heterogeneous.

On the other hand, the birth of digital financial technology was not long ago, but it has been growing exponentially. The development level and direction of digital financial technology in different periods are quite different. Meanwhile, countries all over the world attach increasing importance to ecological efficiency. From the perspective of green innovation and production technology development, urban ecological efficiency worldwide is basically in the rising stage. Therefore, there must be heterogeneity between digital financial technology and ecological efficiency in different periods, and the intensity of their spatial interaction spillover effect may also be heterogeneous. Based on the above analysis, this paper proposes the following hypothesis:

**Hypothesis 4** **(H4).**
*There is temporal and spatial heterogeneity in the intensity of spatial interactive spillover effects between digital financial technology and urban ecological efficiency.*


### 3.2. Methods

According to the primary hypotheses put forward in this paper, the simultaneous equation of the fixed-effect panel regression model is established to verify the simple interactive relationship between digital financial technology and urban ecological efficiency. The basic form of the regression model is as follows:(1)$$e\_{\mathrm{eff}}_{\mathrm{it}}=\alpha_{0}+\alpha_{1}\;d\_{\mathrm{fintech}}_{\mathrm{it}}+{\alpha \;X}_{\mathrm{it}}+\pi_{i}+\varepsilon_{\mathrm{it}}$$
(2)$$d\_{\mathrm{fintech}}_{\mathrm{it}}=\beta_{0}+\beta_{1}\;e\_{\mathrm{eff}}_{\mathrm{it}}+{\beta \;Z}_{\mathrm{it}}+\mu_{i}+\sigma_{\mathrm{it}}\;$$

In Formulas (1) and (2), i represents the individual city and t represents the year; $e\_{\mathrm{eff}}_{\mathrm{it}}$ and $d\_{\mathrm{fintech}}_{\mathrm{it}}$ represent the urban ecological efficiency and the level of digital financial technology, respectively; $X_{\mathrm{it}}$ and $Z_{\mathrm{it}}$ are the control variables that may affect the two variables of each city, including the level of economic development (pgdp), regional urbanization level (urb), industrial structure upgrade (ind), openness (ope), marketization degree (mar), population density (den), transportation development level (tra) and postal development level (pos); $\pi_{i}$ and $\mu_{i}$ represent the controlling individual city; $\varepsilon_{\mathrm{it}}$ and $\sigma_{\mathrm{it}}$ are the error terms.

Simple panel regression models ignore the interaction of variables, while traditional simultaneous equations ignore the possible spatial spillover effect of variables. The traditional spatial econometric models such as spatial lag model (SLM), spatial error model (SEM) and spatial Durbin model (SDM) do not describe the interaction between explanatory variables and explained variables. To investigate the interaction between digital financial technology level and urban ecological efficiency and their spatial spillover effects, this paper constructs the spatial simultaneous equations as follows:(3)$$e\_{\mathrm{eff}}_{\mathrm{it}}=\alpha_{0}+\alpha_{1}\overset{n}{\underset{j\neq i}{\sum }}W\;e\_{\mathrm{eff}}_{\mathrm{it}}+\alpha_{2}\overset{n}{\underset{j\neq i}{\sum }}W\;d\_{\mathrm{fintech}}_{\mathrm{it}}+\alpha_{3}\;d\_{\mathrm{fintech}}_{\mathrm{it}}+{\alpha \;X}_{\mathrm{it}}+\varepsilon_{\mathrm{it}}$$
(4)$$d\_{\mathrm{fintech}}_{\mathrm{it}}=\beta_{0}+\beta_{1}\overset{n}{\underset{j\neq i}{\sum }}W\;d\_{\mathrm{fintech}}_{\mathrm{it}}+\beta_{2}\overset{n}{\underset{j\neq i}{\sum }}W\;e\_{\mathrm{eff}}_{\mathrm{it}}+\beta_{3}\;e\_{\mathrm{eff}}_{\mathrm{it}}+{\beta \;Z}_{\mathrm{it}}+\sigma_{\mathrm{it}}$$

In Formulas (3) and (4), W is the spatial weight matrix. Given the complexity of spatial spillover, this paper constructs the geographic distance spatial weight matrix (W1) and the economic–geographical distance spatial weight matrix (W2), respectively. W1 calculates the linear distance between each other according to the central longitude and latitude coordinates of each city. On the basis of dimensionless processing, the reciprocal value is taken as the weight. If the central distance between two cities exceeds 30, the weight value is 0, that is, the two sample-cities are determined to be non-adjacent; W2 comprehensively considers the economic distance between cities based on geographical distance, and the single element in the matrix is calculated by the following method:(5)$$\mathrm{ECO}\_\mathrm{GEO}\_{\mathrm{Distance}}_{i,j}=\mathrm{GEO}\_{\mathrm{Distance}}_{i,j}\times \mathrm{ECO}\_{\mathrm{Distance}}_{i,j}$$

In Formula (5), $\mathrm{ECO}\_\mathrm{GEO}\_{\mathrm{Distance}}_{i,j}$ represents the economic–geographical distance between city i and city j; $\mathrm{GEO}\_{\mathrm{Distance}}_{i,j}$ represents the geographical distance between city i and city j; $\mathrm{ECO}\_{\mathrm{Distance}}_{i,j}$ represents the absolute value of per capita GDP gap between city i and city j. Then the reciprocal of is also taken as the weight after normalization.

According to the spatial econometric theory, in Formulas (3) and (4), $\alpha_{1}$ represents the spatial spillover intensity and direction of digital financial technology in surrounding cities; $\beta_{1}$ represents the spatial spillover intensity and direction of ecological efficiency of surrounding cities; they are used to test the spatial interaction between digital financial technology and urban ecological efficiency; $\alpha_{2}$ describes the intensity and direction of the impact of ecological efficiency of surrounding cities on local digital financial technology; $\beta_{2}$ describes the intensity and direction of the impact of digital financial technology of surrounding cities on local urban ecological efficiency, and they are used to describe the endogenous relationship between digital financial technology and urban ecological efficiency

Based on the above spatial simultaneous equations, the GS3SLS method is used for holistic estimation. Since the digital financial technology and the urban ecological efficiency in the spatial simultaneous equation model are endogenous variables, the OLS estimation will lead to the loss of consistency of the estimation results; besides, the spatial simultaneous equations will lead to over-recognition according to the condition of model recognition, so the GS3SLS method is suitable to estimate the spatial simultaneous equations holistically. The GS3SLS method takes into account the potential spatial correlation of endogenous variables and the possible correlation between random disturbance terms of each equation, which improves the effectiveness of the estimation results.

### 3.3. Variable Description and Data Sources

Based on the data availability, 284 prefectural and above cities in China are selected as samples, containing 3124 samples from 2008 to 2018. The main variables are digital financial technology level (d-fintech) and Urban ecological efficiency level (e-eff). d-fintech data are obtained from the information search of Baidu (baidu.com, accessed on 16 March 2021) through Python network crawler technology, and e-eff data are calculated by the DEA model based on super-efficiency SBM-GML. The control variables are: economic development level (pgdp), calculated by per capita GDP of each city and taking logarithm; regional urbanization level (urb), calculated with the proportion of the number of people in municipal districts in the total population of the city; industrial structure upgrading (ind), calculated with the proportion of added value of the city’s secondary and tertiary industries in GDP; openness (ope), represented by the proportion of the city’s actual foreign investment in GDP; marketization degree (mar), measured with the relative proportion of the number of people employed in private and individual units in the total population; population density (den), measured with population per square kilometer of the whole city and taking logarithm; transport development level (tra), calculated by the total passenger volume of the city divided by the total population and taking logarithm; development level of posts (pos), measured by the city’s postal business income divided by the total population and taking logarithm. The data of control variables are obtained from China City Statistical Yearbooks, and some missing values are completed by interpolation according to the changing trend. GeoDa and Stata16 software are used for data processing. Table 1 reports data sources for all variables in this paper.

Table 2 reports descriptive statistics for all variables in this paper, presenting descriptive statistics of cities in eastern, central and western regions. To reflect the period heterogeneity, the descriptive statistics of sample cities in 2008–2012 and 2013–2018 are presented, respectively. It can be seen that all variables of sample cities are different from different angles. Based on this, this paper will analyze the heterogeneity of spatial interaction spillovers between digital financial technology level and urban ecological efficiency based on the differences of various dimensions in the fifth section.

### 3.4. Pretreatment of Variables

#### 3.4.1. Measurement of Digital Financial Technology

Generally speaking, there are two main ways to obtain external data: The first is to obtain externally public data sets. For example, some scientific research institutions, enterprises, and governments will release some data. These data sets are usually relatively complete and relatively high quality. Currently, there are no relevant data on digital financial technology at the city level in China. The second is to use crawler tools to crawl the Internet, such as obtaining recruitment information for a certain position from a recruitment website, rental house websites to obtain rental information in a certain area, and e-commerce websites to obtain information about a certain product, etc. Based on this we can perform data analysis on the crawled data.

There are few quantitative studies on digital financial technology indicators in the past. In this paper, we refer to Yao et al. [47] and extract keywords related to digital financial technology according to “The 13th Five-Year National Science and Technology Innovation Plan”, “Big Data Industry Development Plan (2016–2020)”, “China Fintech Operation Report (2018)” and relevant important news and conferences.

There are 48 keywords in total, including: EB level storage, NFC payment, differential privacy technology, big data, third-party payment, multi-party secure computing, distributed computing, equity crowdfunding financing, Internet finance, machine learning, open banking, brain-like computing, quantitative finance, flow computing, green computing, memory computing, blockchain, artificial intelligence, cognitive computing, fusion architecture, business intelligence, authentication deep learning, biometric technology, data visualization, data mining, digital currency, investment decision support system, graph computing, image understanding, Internet connection, text mining, Internet of things, information physics system, virtual reality, mobile Internet, mobile payment, 100 million level concurrency, heterogeneous data, semantic search, speech recognition, cloud computing, credit investigation, intelligent financial contract Intelligent customer service, intelligent data analysis, intelligent investment consultant and natural language processing.

Then we match these keywords with the sample cities and search them on the “Baidu Information” web page. The Python web crawler technology is used to crawl the web page source code of the “Baidu Information” web page and extract the number of search results. We add up the search results numbers of all keywords of the same city to obtain the total search volume and take logarithm to alleviate heteroscedasticity. The final value is used as an index to measure a sample city’s digital financial technology development level. Figure 3 reports the average distribution of sample cities’ digital financial technology levels during the investigation period (the figure only shows sample cities of this research, excluding sea areas, etc.).

#### 3.4.2. Measurement of Urban Ecological Efficiency

DEA (Data Envelopment Analysis) is the most commonly used method to measure ecological efficiency in previous studies. In this paper, urban ecological efficiency is measured using the DEA approach based on the super-efficiency SBM-GML model [48,49]. DEA was first proposed in 1978 to evaluate the relative efficiency of a group of decision-making units with multiple inputs and outputs [50]. The distance functions of the baseline model are CCR and BCC models, but they do not consider the “Slack” phenomenon. To make up for this shortcoming, Tone put forward the SBM model and the super-efficiency SBM model in 2001 and 2002, respectively. The latter not only considers the relaxation variable but also can rank the decision-making units with the efficiency value greater than 1 [51,52]. The Malmquist-TFP index was first introduced by Malmquist [53] and formally developed in Caves’ innovative research [54]. It is used to measure the TFP change between two periods, and the directional distance function containing undesired output is introduced into the Malmquist index to support the analysis of undesired output. To facilitate intertemporal comparison and overcome the problem of no viable solution, Oh included the production unit in the Global reference set and constructed the Global-Malmquist-Luenberger (GML) index [55].

Based on the above DEA distance function and panel data model, this paper selects the social fixed asset investment, employees, built-up area and energy consumption of the sample cities as the input indicators. Take GDP as the expected output and represent the total output value; waste water, waste gas and smoke emissions are regarded as unexpected outputs, representing pollution emissions. The specific input–output indicators are shown in Table 3. Figure 4 reports the average distribution of ecological efficiency levels of sample cities during the investigation period (the figure only shows sample cities of this research, excluding sea areas, etc.).

## 4. Econometric Examination of the Spatial Interaction Spillover Effects of Digital Financial Technology and Urban Ecological Efficiency

### 4.1. Results of Parameter Estimation

In this paper, the fixed-effect panel regression model Equations (1) and (2) are used to analyze the interaction between digital financial technology and urban ecological efficiency. According to the model setting, parameter estimation results are shown in Table 4.

According to Table 4, there is a significant interaction between digital financial technology and urban ecological efficiency. Columns (1) and (4) represent situations in which no individual is controlled, and control variables are not considered; Columns (2) and (5) represent situations in which no individual is controlled, but control variables are considered; Columns (3) and (6) represent situations in which both individuals are controlled, and control variables are considered. It can be seen from Columns (1)–(3) that urban ecological efficiency promotes digital financial technology. According to Columns (4)–(6), digital financial technology also plays a role in promoting urban ecological efficiency.

The simple panel regression does not consider the spatial relationship between variables. According to the spatial simultaneous Equations (3) and (4) and the GS3SLS estimation method, the spatial interaction spillover relationship between digital financial technology and urban ecological efficiency is calculated [56,57,58]. According to the setting, the parameters of the models based on geographical distance spatial weight and economic–geographical distance spatial weight are estimated, respectively, and the results are shown in Table 5.

According to Table 5, the parameter estimation results of geographical distance spatial weight and economic–geographical distance spatial weight are basically consistent in direction. The results are as follows: the parameter estimation results of Columns (1) and (3) show that the digital financial technology of surrounding cities has a negative impact on the ecological efficiency of local cities; the urban ecological efficiency of surrounding cities promotes the ecological efficiency of local cities; the level of local digital financial technology promotes the ecological efficiency of local cities. The parameter estimation results of Columns (2) and (4) show that the digital financial technology of surrounding cities promotes the local digital financial technology; the urban ecological efficiency of surrounding cities has a negative impact on local digital financial technology, but the negative impact is not significant from the perspective of economic–geographical distance; local urban ecological efficiency promotes local digital financial technology. On the premise that the significance is basically passed, and the economic meanings are the same, the coefficients can be compared. From the perspective of pure geographical distance, the inhibitory effect of digital financial technology in surrounding cities on the ecological efficiency of local cities is greater than the promoting effect of local digital financial technology on local ecological efficiency; the inhibitory effect of ecological efficiency in surrounding cities on digital financial technology in local cities is greater than the promoting effect of local ecological efficiency on local digital financial technology.

### 4.2. Analysis of Empirical Results

#### 4.2.1. General Interaction Effects of Digital Financial Technology and Urban Ecological Efficiency

Through parameter estimation, it is found that digital financial technology and urban ecological efficiency have mutually promoting effects. From the perspective of pure geographic distance, for every unit that the level of digital financial technology is improved, the urban ecological efficiency will be significantly improved by 0.489. The level of digital financial technology will significantly increase by 1.936 for every unit of urban ecological efficiency improvement. On the whole, there is an interactive effect between digital financial technology and urban ecological efficiency, and the marginal effect of urban ecological efficiency on the level of digital financial technology is more prominent, which is in a relatively dominant position in the mutual promotion relationship between the two. The improvement of urban ecological efficiency means the intensive utilization of urban development resources, and the new technology and system behind it have significant adsorption on digital finance. It can also provide a better industrial foundation and market potential for the development of the financial industry and promote the innovative development of digital finance.

From the perspective of economic–geographical distance, the urban ecological efficiency is significantly improved by 0.345 for each unit of digital financial technology improvement. For every unit of urban ecological efficiency improvement, the level of digital financial technology will be significantly improved by 0.814. The conclusion of the general interaction effect under the perspective of economic–geographical distance is consistent with that under the perspective of pure geographical distance.

#### 4.2.2. Spatial Spillover Effects of Digital Financial Technology and Urban Ecological Efficiency

Through parameter estimation, it is found that both digital financial technology and urban ecological efficiency have significant spatial spillovers. From the perspective of geographical distance, the level of local digital financial technology is significantly increased by 1.206 for every unit of improvement in digital financial technology in the surrounding cities. The digital financial technology in the surrounding cities can positively impact the development of local digital financial technology through the external expansion radiation of financial enterprises and the common market effect. Each unit of improvement in the ecological efficiency of the surrounding cities can significantly improve the ecological efficiency of the local cities by 1.263. The improvement of the ecological efficiency of the surrounding cities can improve the ecological efficiency of the local cities through innovation, demonstration and knowledge spillover effect.

From the perspective of economic–geographical distance, the local digital financial technology will increase significantly by 0.112 for each unit of improvement in the digital financial technology of surrounding cities with similar development levels. For each unit of increase in the ecological efficiency of surrounding cities with similar development levels, the ecological efficiency of local cities will be significantly increased by 0.691. The conclusion of spatial spillover effect under economic–geographical distance is consistent with that under the perspective of geographical distance.

#### 4.2.3. Spatial Interaction Effects of Digital Financial Technology and Urban Ecological Efficiency

Through parameter estimation, this paper found that the digital financial technology of surrounding cities has a restraining effect on the ecological efficiency of local cities, and the improvement of the ecological efficiency of surrounding cities has a siphon effect on the local digital financial technology. From the perspective of geographical distance, the ecological efficiency of local cities is significantly reduced by 0.605 for every unit of improvement of the digital financial technology of surrounding cities. As one of the driving factors of regional innovation and development, the spread of digital financial technology to surrounding cities weakens the local development momentum. The cities with lagging digital finance development should consolidate their own development resources, strengthen the endogenous support mechanism of local digital financial technology to urban ecological efficiency, and resist the inhibition effect from surrounding cities. For each unit of ecological efficiency improvement in surrounding cities, the level of local digital financial technology has been significantly reduced by 2.444. The improvement of ecological efficiency in surrounding cities can lead to the optimization of the industrial development environment and innovation of ecosystem development, which can generate a strong siphon effect on local digital financial technology. Therefore, cities with low ecological efficiency levels need to strengthen their own industrial progress to curb the outflow of digital financial technology.

From the perspective of economic–geographical distance, the ecological efficiency of local cities decreases significantly by 0.131 for each unit increase in the digital financial technology of surrounding cities with similar development levels. The conclusion is consistent with that from the perspective of geographical distance. When the ecological efficiency of surrounding cities with similar development levels increases by one unit, the local digital financial technology level decreases by 0.131, but it is not significant, indicating that in fact, the siphon effect of ecological efficiency is not obvious among cities at the same development level, and the siphon effect is highly related to the economic development level of the city itself, so blindly improving ecological efficiency while ignoring economic development cannot have a strong attraction to external digital financial technology.

### 4.3. Robustness Test

On the basis of the results of the spillover effect test, the robustness test is conducted in this part. The effects of the adjusted distance band width, the adjusted spatial weighting matrix type and the adjusted variable measurement model on the robustness of the analysis conclusions are investigated.

#### 4.3.1. Robustness Test of Distance Band

For the theoretical model with the addition of spatial correlation analysis, the selection of the distance band determines the number of “neighbors” of the sample city, so the setting of the distance band may affect the test results of spatial effect. To test whether the adjustment of distance band affects the robustness of the model analysis, the models shown in Table 6 reduce the distance band from the benchmark W: 0–30 to W: 0–20 and extend it to W: 0–40, respectively, and then the sample city will have fewer or more neighbors in the spatial matrix.

From Table 6, we can see that the regression results are still robust. Adjusting the setting of the distance band will not affect the parameter estimation results. First of all, digital financial technology and urban ecological efficiency promote each other, and urban ecological efficiency is in a comparatively dominant position. Secondly, both digital financial technology and urban ecological efficiency have significant spatial spillovers. Finally, digital financial technology in surrounding cities has a restraining effect on the ecological efficiency of local cities, and the improvement of ecological efficiency in surrounding cities has a siphon effect on the local digital financial technology.

#### 4.3.2. Robustness Test of Contiguity Spatial Weighting Matrix

Spatial weighting matrix is a crucial parameter of spatial panel data model. The setting method of spatial weighting matrix may affect the test results of spatial effect. In this paper, the numerical distance weighting matrix is originally used to represent the closeness of the relationship between sample cities. In this section, the adjacent distance weighting matrix is used for the robustness test; that is, the adjacent unit distance is 1 and the non-adjacent unit distance is 0. The adjusted spatial weighting matrix combined with different distance band settings is used to investigate the spatial effect between digital financial technology and urban ecological efficiency. The analysis results are shown in Table 7.

By observing Table 7, it can be found that the regression results are still robust. Adjusting the setting method of the spatial weighting matrix will not affect the parameter estimation results. First of all, digital financial technology and urban ecological efficiency promote each other, and urban ecological efficiency is in a comparatively dominant position. Secondly, both digital financial technology and urban ecological efficiency have significant spatial spillovers. Finally, digital financial technology in surrounding cities has a restraining effect on the ecological efficiency of local cities, and the improvement of ecological efficiency in surrounding cities has a siphon effect on the local digital financial technology.

#### 4.3.3. Robustness Test of Adjusted Variables

The above urban ecological efficiency is calculated by using the DEA model based on super-SBM-GML. In this section, two methods are used to adjust the calculation of urban ecological efficiency, so as to avoid the possibility of a specific calculation model leading to the results of this paper, similar to a placebo test. First, this section uses the input–output data of the expanded year (2003–2018) to calculate the original model. Since the GML model is the most effective frontier method for comprehensive consideration, the results obtained after expanding the sample year are more holistic. Second, this section uses the static global-super-SBM model to calculate the urban ecological efficiency; that is, the original panel model GML is changed to a common panel model with 11 windows. Compared with the original dynamic GML value, this result is no longer the ratio multiplication between years, more directly reflecting the ecological efficiency value of the sample city in a specific year. In this section, the urban ecological efficiency values calculated by the above two methods are used for the robustness test, and other settings are consistent with the above. The analysis results are shown in Table 8.

By observing Table 8, it can be found that the regression results are still robust, and adjusting the setting of the measurement model of urban ecological efficiency will not affect the parameter estimation results, which are consistent with the original results.

In summary, the conclusions obtained by using the spatial simultaneous equations and the GS3SLS estimation method are robust and effective. This paper accepts hypotheses H1, H2, and H3.

## 5. Spatial–Temporal Heterogeneity of the Spatial Interaction Spillover Effects

This section will test the spatial–temporal heterogeneity of the spatial interaction spillover effect between digital financial technology and urban ecological efficiency. The level of digital financial technology and urban ecological efficiency show apparent temporal and spatial heterogeneity in descriptive statistics. Specifically, the level of digital financial technology in the economically developed eastern region is higher, and the level of digital financial technology in 2013–2018 is also significantly higher than that in 2008–2012; the urban ecological efficiency in the central region is slightly higher than that in the eastern and western regions, and the urban ecological efficiency in 2013–2018 is significantly higher than that in 2008–2012.

### 5.1. Heterogeneity Analysis Based on the Regional Differences

According to different economic and social development levels, China can be divided into three economic zones, based on which this paper divides sample cities into eastern, central and western cities. Following Equations (3) and (4), the spatial simultaneous equations and the GS3SLS estimation method are used in this section to analyze the regional heterogeneity of the spatial interaction spillovers between digital financial technology and urban ecological efficiency. The parameter estimation results are shown in Table 9 and Table 10.

According to the estimation results of Table 9 and Table 10, there is regional heterogeneity in the spatial interaction spillover effect between digital financial technology and urban ecological efficiency.

#### 5.1.1. Analysis of General Interaction Effects Based on the Regional Heterogeneity

The regional heterogeneity of the general interaction effect is analyzed in this part. From the perspective of geographical distance, the impact coefficients of ecological efficiency on digital financial technology in eastern, central and western cities are 1.599, 0.915, and 1.518, respectively, with the central cities in a comparatively weak position. Eastern cities have inherent advantages in economic development and a strong development capacity of digital financial technology. With the implementation of industrial transfer policies and inclined regional development policies in recent years, cities in the western region have obtained a late-starter advantage, and their urban ecological efficiency has significantly improved the ability to absorb digital financial technology. The impact coefficients of digital financial technology on ecological efficiency in eastern, central and western cities are 0.279, 0.728, and 0.653, respectively, with the eastern cities in a relatively weak position. The main reason is that the level of digital financial technology in eastern cities has improved too fast. Although ecological efficiency has been promoted, the speed of improvement cannot keep up with the rapid changes of digital financial technology in the eastern region; moreover, the economic and industrial volume of eastern cities are large, and the difficulty of significantly improving ecological efficiency is greater than that in central and western cities.

From the perspective of economic–geographical distance, the impact coefficients of ecological efficiency on digital financial technology in eastern, central and western cities are 1.581, 1.102, and 1.290, respectively, with the central cities in a comparatively weak position. The impact coefficients of digital financial technology on ecological efficiency in eastern, central and western cities are 0.609, 0.708 and 0.723, respectively, with the eastern cities in a relatively weak position. In general, there are differences in the intensity of interaction between digital financial technology and urban ecological efficiency in eastern, central and western cities.

#### 5.1.2. Analysis of Spatial Spillover Effects Based on the Regional Heterogeneity

The regional heterogeneity of spatial spillover effect is analyzed in this part. From the perspective of geographical distance, the impact coefficients of the ecological efficiency of the surrounding cities on the local ecological efficiency in the eastern, central and western regions are 1.059, 1.191, and 1.196, respectively, and the intensity is basically the same, indicating that the positive spatial spillover effect of urban ecological efficiency in each region is noticeable and the intensity is the same. The impact coefficients of digital financial technology in the surrounding cities on local digital financial technology in the eastern, central and western regions are 0.876, 1.034, and 0.968, respectively, and the intensity is basically the same, indicating that the positive spatial spillover effect of digital financial technology in each region is obvious and the intensity is the same.

From the perspective of economic–geographical distance, the impact coefficients of the ecological efficiency of the surrounding cities with similar development levels on the local ecological efficiency in the eastern, central and western regions are 11.573, 64.775, and 18.941, respectively, indicating that the ecological efficiency spillover effect among cities with similar development level is stronger in the central region. The impact coefficients of digital financial technology on local digital financial technology in surrounding cities with similar development level in the eastern, central and western regions are 3.360 (insignificant), 14.637, and 7.527, respectively, indicating that the spillover effect of digital financial technology among cities with similar development level is more intense in the central region. However, the cooperation effect of financial technology level among cities with similar development level in the eastern region is not significantly stronger than the competition effect.

#### 5.1.3. Analysis of Spatial Interaction Effects Based on the Regional Heterogeneity

This part analyzes the regional heterogeneity of the spatial interaction effects. From the perspective of geographical distance, the impact coefficients of ecological efficiency of surrounding cities on local digital financial technology in the eastern, central and western regions are −1.776, −1.134, and −1.821, respectively, and the improvement of ecological efficiency in surrounding cities in the central region has the smallest siphon effect on local digital financial technology. Combined with the standard deviation statistics of digital financial technology (0.915 in eastern cities, 0.703 in central cities and 0.703 in western cities) shown in Table 2, it can be found that the layout of digital finance in the eastern and western regions are more unbalanced, making the improvement of urban ecological efficiency have a stronger adsorption capacity for digital financial technology. The impact coefficients of digital financial technology of surrounding cities on local urban ecological efficiency in eastern, central, and western China are −0.299, −0.822, and −0.634, respectively. Combined with the average values of the digital financial technology shown in Table 2 (4.636 in eastern cities, 4.227 in central cities and 4.286 in western cities), it can be found that in regions with higher overall digital financial level, the inhibitory effect of digital financial technology in surrounding cities on local ecological efficiency is smaller.

From the perspective of economic–geographical distance, the impact coefficients of ecological efficiency of surrounding cities with similar development level on local digital financial technology in the eastern, central and western regions are −12.526 (not significant), −37.579, and −20.599, respectively, indicating that the siphon effect of ecological efficiency on digital financial technology among cities with similar development level is stronger in the central region, while the siphon effect is not significant in the eastern region. The impact coefficients of digital financial technology of surrounding cities with similar development level on local urban ecological efficiency in the eastern, central and western regions are −3.384 (insignificant), −23.996, and −6.770, respectively, which present consistent siphon effect of urban ecological efficiency on digital financial technology. In general, the spatial interaction between ecological efficiency and digital financial technology is not significant among cities with similar development levels in the economically developed eastern region. However, in the cities with the same level of development in central and western China, the spatial interaction between the two has significant inhibition and siphon effects, indicating that the competition effect is greater than the cooperation effect between cities with the same level of development in central and western China.

### 5.2. Heterogeneity Analysis Based on the Period Differences

It is generally believed that China’s digital financial technology level experienced two development stages around 2013. First, in 2013, the Bank of China took the lead in releasing the “Bank of China Open Platform” in China to provide digital financial services and data mining services for partners and customers, leading other financial enterprises to launch open banking platforms or businesses one after another. Secondly, Alipay, under Alibaba, launched its “Yu Ebao” business in 2013, which greatly increased the participation of ordinary residents in digital financial business. Since then, various funds and insurance companies have launched a large-scale Internet-based strategic layout, and Internet financial enterprises have achieved unprecedented development due to their advantages in terms of experience and technical capabilities. Following Equations (3) and (4), the spatial simultaneous equations and the GS3SLS estimation method are used in this section to analyze the temporal heterogeneity of the spatial interaction spillovers between digital financial technology and urban ecological efficiency. The parameter estimation results are shown in Table 11 and Table 12.

According to the estimation results in Table 11 and Table 12, the spatial interaction spillover effect between digital financial technology and urban ecological efficiency is heterogeneous in different periods.

#### 5.2.1. Analysis of General Interaction Effects Based on the Period Heterogeneity

This sub-section analyzes the period heterogeneity of the general interaction effect. From the perspective of geographical distance, the impact coefficients of urban ecological efficiency on digital financial technology during 2008–2012 and 2013–2018 are 1.641 and 1.125, respectively. The promoting effect of urban ecological efficiency on digital financial technology shows a downward trend as time goes by. From 2008 to 2012 and from 2013 to 2018, the impact coefficients of urban digital financial technology on urban ecological efficiency are 0.370 and 0.536, respectively. The promoting effect of digital financial technology on urban ecological efficiency is on the rise as time goes by. In other words, the innovative effect of digital financial level on urban green development has been strengthened. It is necessary to strengthen further the role of digital finance in promoting urban ecological efficiency.

From the perspective of economic–geographical distance, the impact coefficients of urban ecological efficiency on digital financial technology in 2008–2012 and 2013–2018 are 1.532 and 0.884, respectively. The impact coefficients of urban digital financial technology on urban ecological efficiency in 2008–2012 and 2013–2018 are 0.480 and 0.522, respectively. In summary, the general interaction between digital financial technology and urban ecological efficiency from the economic–geographical perspective has the same trend in different periods as that from the pure geographical perspective.

#### 5.2.2. Analysis of Spatial Spillover Effects Based on the Period Heterogeneity

The period heterogeneity of the spatial spillover effect is analyzed in this sub-section. From the perspective of geographical distance, the impact coefficients of ecological efficiency of surrounding cities on local ecological efficiency during 2008–2012 and 2013–2018 are 1.030 and 1.251, respectively. The sharing of innovative resources among regions strengthened the spillover effect of urban ecological efficiency to a certain extent. From 2008 to 2012 and from 2013 to 2018, the impact coefficients of digital financial technology in surrounding cities on local digital financial technology are 0.980 and 0.987, respectively. The spillover effects of digital financial technology are basically the same in the two periods.

From the perspective of economic–geographical distance, the impact coefficients of ecological efficiency of surrounding cities with similar development levels on local ecological efficiency during 2008–2012 and 2013–2018 are 0.260 and 0.607, respectively, which are consistent with the changing trend under the geographical perspective. From 2008 to 2012 and from 2013 to 2018, the impact coefficients of digital financial technology of surrounding cities with similar development levels on local digital financial technology are 0.141 (insignificant) and 0.032 (insignificant), respectively, indicating that the cooperation effect of financial technology level between cities with similar development level in the two periods is not significantly stronger than the competition effect.

#### 5.2.3. Analysis of Spatial Interaction Effects Based on the Period Heterogeneity

This sub-section analyzes the period heterogeneity of spatial interaction effect. From the perspective of geographical distance, the impact coefficients of ecological efficiency of surrounding cities on local digital financial technology during 2008–2012 and 2013–2018 are −1.753 and −1.361, respectively, indicating that the overall improvement of urban ecological efficiency leads to a decline in the spatial siphon effect on digital financial technology, which means that the spatial equilibrium of digital financial development needs to be further improved. From 2008 to 2012 and from 2013 to 2018, the impact coefficients of digital financial technology in surrounding cities on local urban ecological efficiency are −0.376 and −0.657, respectively. The downward pressure on local ecological efficiency caused by inter-regional digital financial technology competition has increased.

From the perspective of economic–geographical distance, the impact coefficients of ecological efficiency of surrounding cities with similar development levels on local digital financial technology in 2008–2012 and 2013–2018 are 0.132 (insignificant) and −0.039 (insignificant), respectively, indicating that the siphon effect of ecological efficiency on digital financial technology is not obvious between cities with similar development levels in the two periods. The impact coefficients of digital financial technology in surrounding cities with similar development levels on local urban ecological efficiency in 2008–2012 and 2013–2018 are −0.118 and −0.108, respectively. The spatial inhibitory effect of digital financial technology on local urban ecological efficiency is basically the same in the two periods.

In summary, the spatial interaction spillover between digital financial technology and urban ecological efficiency does have temporal and spatial heterogeneity. This paper accepts hypothesis H4.

## 6. Conclusions and Implications

There is a complex interactive mechanism between digital financial technology and urban ecological efficiency, and the spillover effects of their spatial interaction need to be explored deeply due to the special spatial distribution characteristics. In this paper, Python web crawler technology and the super-efficiency SBM-GML model are used to obtain the financial technology level index and urban ecological efficiency index of 284 cities in China from 2008 to 2018, respectively. Based on the spatial simultaneous equations and the GS3SLS estimation method, the research draws the following conclusions.

First, there are spatial spillover effects between digital financial technology and urban ecological efficiency. (1) There is a mutual promotion effect between digital financial technology and urban ecological efficiency, and urban ecological efficiency occupies a dominant position. (2) From the perspective of spatial spillover, both digital financial technology and urban ecological efficiency have significant spatial spillover effects. (3) From the perspective of spatial interaction, digital financial technologies in surrounding cities have a restraining effect on the ecological efficiency of local cities, and the improvement of ecological efficiency in surrounding cities has a siphon effect on local digital financial technology. All the above conclusions passed the robustness test of the adjusted distance band, adjusted space weighting matrix and adjusted variable measurement model.

Second, the spatial interaction between digital financial technology and urban ecological efficiency has regional heterogeneity in the intensity of spillover effect. (1) In the general interaction effect analysis, the central region shows a comparatively weak trend in promoting urban ecological efficiency to digital financial technology, and the eastern region shows a weak trend in the promotion of digital financial technology to ecological efficiency. (2) In the analysis of the spatial spillover effect, the spatial spillover intensity of both digital financial technology and urban ecological efficiency in each region is basically the same. (3) In the analysis of spatial interaction effect, the improvement of ecological efficiency of surrounding cities in the central region has less siphon effect on local digital financial technology, and the digital financial technology of surrounding cities in the eastern region has a less inhibitory effect on local ecological efficiency.

Finally, the spatial interaction between digital financial technology and urban ecological efficiency has period heterogeneity in the intensity of spillover effect. (1) In the general interaction effect analysis, the promotion effect of urban ecological efficiency on digital financial technology shows a downward trend over time, and the promotion effect of digital financial technology on urban ecological efficiency shows an upward trend over time. (2) In the analysis of the spatial spillover effect, the sharing of innovation resources among regions strengthens the spillover effect of urban ecological efficiency to a certain extent, while the spillover effect of digital financial technology is basically the same in the two periods. (3) In the analysis of the spatial interaction effect, the overall improvement of urban ecological efficiency leads to the decline of the spatial siphon effect on digital financial technology, and the downward pressure on local ecological efficiency due to regional digital financial technology competition is enhanced.

The conclusions of this paper can be used as a reference for the development policies of digital financial technology and ecological efficiency. First, perfecting the systematic driving force of urbanization to improve urban ecological efficiency. Urban ecological efficiency plays a dominant role in the mutual promotion relationship with digital financial technology. Therefore, from the perspective of regional digital financial technology competition, it is an alternative strategy for attracting financial and innovative factors by improving urban ecological efficiency. Second, the regional financial ecosystem should be improved according to local conditions to improve the efficiency and utilization performance of digital financial technology resource allocation. As the core of the modern economy, the financial industry plays a vital role in regional economic growth. Therefore, it is necessary to strengthen the construction of financial infrastructure and build a multi-level financial development format. Third, the mutual promotion relationship between digital financial technology and urban ecological efficiency should be strengthened to optimize the momentum of urban development. Full play should be given to the allocation and baton role of financial resources to improve the financial support for urban innovative and green development. The spillover effect of digital financial technology should be enhanced to promote urban ecological efficiency through green finance. In general, countries around the world need to implement domestic and foreign policies regarding aspects like digital finance, environmental protection, science and technology and the supply side according to their own conditions, gradually improve the level of digital financial technology and its green development capacity, and steadily improve their long-term development mechanism on the premise of ensuring production capacity.

## Figures and Tables

**Figure 1 ijerph-18-08535-f001:**
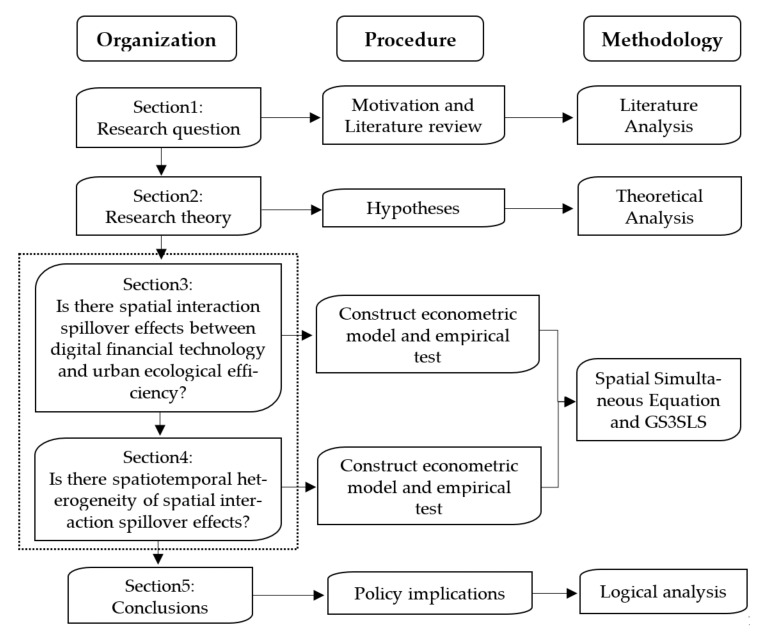
The logical framework of this paper.

**Figure 2 ijerph-18-08535-f002:**
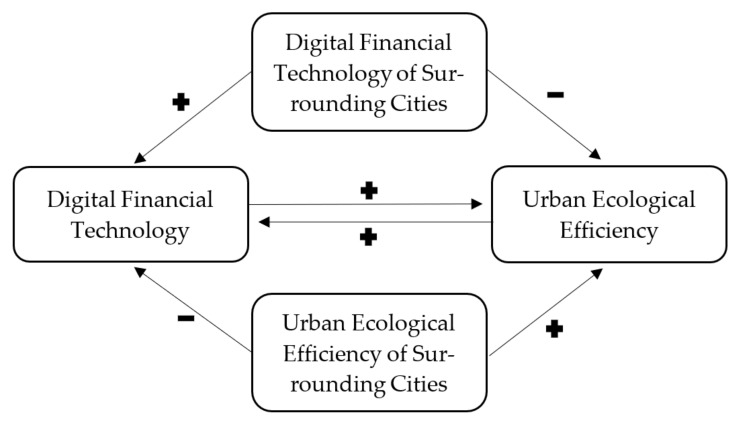
Interactive mechanism of digital financial technology and urban ecological efficiency.

**Figure 3 ijerph-18-08535-f003:**
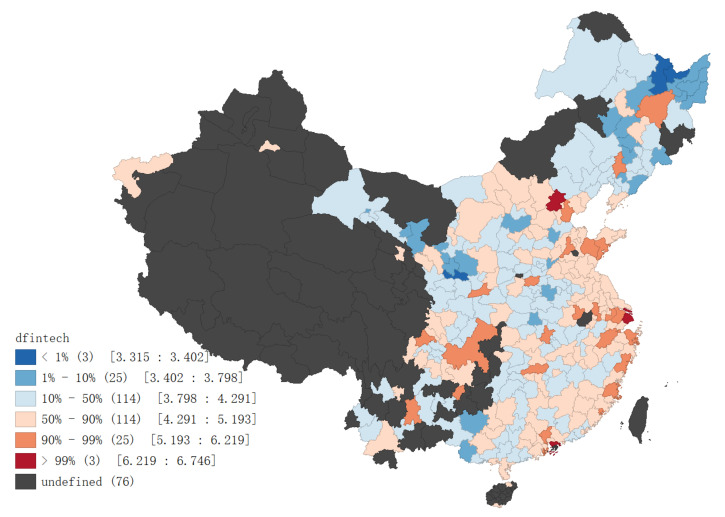
Mean d-fintech values from 2008 to 2018.

**Figure 4 ijerph-18-08535-f004:**
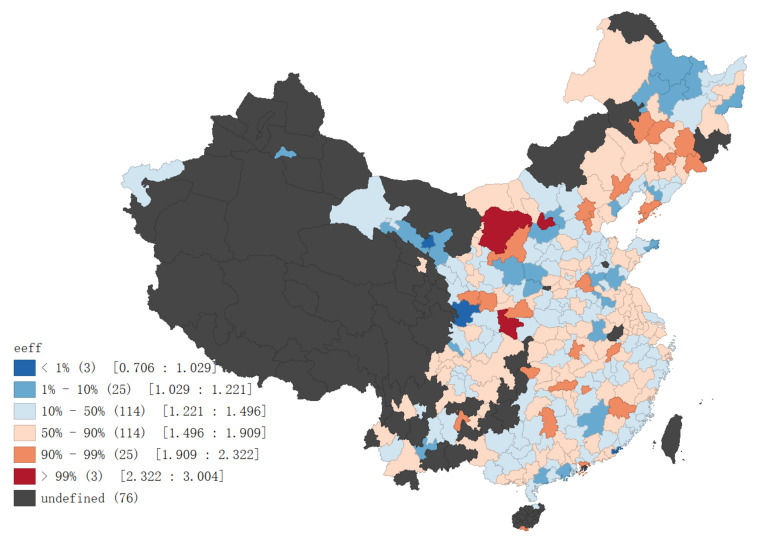
Mean e-eff values from 2008 to 2018.

**Table 1 ijerph-18-08535-t001:** Variable data sources.

Variable	Abbr.	Source
Digital Financial Technology	d-fintech	Web Crawler Search
Urban Ecological Efficiency	e-eff	Data Envelopment Analysis
Economic Development Level	pgdp	China City Statistical Yearbook
Regional Urbanization Level	urb	China City Statistical Yearbook
Upgrading of Industrial Structure	ind	China City Statistical Yearbook
Openness	ope	China City Statistical Yearbook
Degree of Marketization	mar	China City Statistical Yearbook
Population density	den	China City Statistical Yearbook
Transportation Development Level	tra	China City Statistical Yearbook
Development Level of Posts	pos	China City Statistical Yearbook

**Table 2 ijerph-18-08535-t002:** Descriptive Statistics.

	Unit	Items	Summary	Eastern Region	Central Region	Western Region	2008–2012	2013–2018
N	-	-	3124	1232	1221	671	1420	1704
d-fintech	-	Mean	4.401	4.636	4.227	4.286	3.974	4.757
		Std	0.833	0.915	0.703	0.793	0.587	0.841
e-eff	-	Mean	1.528	1.516	1.553	1.504	1.256	1.754
		Std	0.516	0.464	0.535	0.57	0.314	0.542
pgdp	ln(RMB/person)	Mean	10.498	10.725	10.428	10.211	10.223	10.728
		Std	0.644	0.611	0.576	0.676	0.634	0.556
urb	ratio	Mean	0.357	0.388	0.321	0.364	0.341	0.369
		Std	0.238	0.261	0.215	0.223	0.237	0.237
ind	ratio	Mean	0.872	0.893	0.86	0.854	0.862	0.879
		Std	0.081	0.068	0.088	0.081	0.084	0.077
ope	ratio	Mean	0.027	0.027	0.038	0.008	0.036	0.02
		Std	0.096	0.059	0.138	0.04	0.138	0.03
mar	ratio	Mean	0.128	0.167	0.111	0.088	0.1	0.151
		Std	0.136	0.188	0.082	0.067	0.112	0.15
den	ln(person/km^2^)	Mean	5.725	6.109	5.597	5.256	5.706	5.742
		Std	0.917	0.672	0.951	0.959	0.909	0.923
tra	ln(RMB/person)	Mean	2.771	2.799	2.678	2.891	2.944	2.628
		Std	0.784	0.863	0.692	0.77	0.765	0.771
pos	ln(RMB/person)	Mean	4.235	4.424	4.155	4.034	3.935	4.485
		Std	0.796	0.868	0.731	0.693	0.69	0.793

**Table 3 ijerph-18-08535-t003:** Measurement of input and output indicators by urban ecological efficiency.

First-Grade Indicators	Second-Grade Indicators	Third-Grade Indicators
Input indicator	Capital	Investment in fixed assets
	Labor	Employees
	Construction	Built up area
	Energy	Energy consumption
Output indicator	Desirable output	GDP
	Undesirable output	Waste water
		Waste gas
		Smoke and dust

**Table 4 ijerph-18-08535-t004:** Results of benchmark regression without spatial effect.

	Dependent Variable: d-fintech			Dependent Variable: e-eff		
	(1)	(2)	(3)	(4)	(5)	(6)
e-eff	0.720 *** (27.85)	0.352 *** (16.23)	0.247 *** (12.54)	-	-	-
d-fintech	-	-	-	0.277 *** (27.85)	0.221 *** (16.23)	0.213 *** (12.54)
pgdp	-	0.453 *** (16.14)	0.921 *** (27.81)	-	0.276 *** (12.20)	0.511 *** (15.33)
urb	-	−0.001 (−0.02)	0.825 *** (7.79)	-	−0.295 *** (−7.09)	0.511 *** (−2.98)
ind	-	0.136 (0.71)	−1.456 *** (−4.10)	-	−0.749 *** (−4.94)	0.371 (1.12)
ope	-	−0.205 ** (−2.02)	−0.059 (−0.84)	-	0.174 ** (2.17)	0.181 *** (2.76)
mar	-	1.155 *** (12.03)	1.607 *** (14.52)	-	−0.241 *** (−3.11)	−0.271 ** (−2.55)
den	-	0.202 *** (17.50)	0.252 *** (4.18)	-	−0.066 *** (−6.95)	−0.177 *** (−3.16)
tra	-	−0.108 *** (−7.81)	−0.141 *** (−10.12)	-	−0.067 *** (−6.09)	−0.074 *** (−5.64)
pos	-	0.146 *** (9.23)	0.119 *** (6.79)	-	0.061 *** (4.79)	0.145 *** (8.93)
Cons	3.302	−2.631	−6.430	0.311	−1.254	−4.355
FE	No	No	Yes	No	No	Yes
N	3124	3124	3124	3124	3124	3124
R2	0.1990	0.5718	0.7044	0.1990	0.2996	0.5391
F	775.79	462.08	749.64	775.79	148.01	367.98

Notes: **, *** stand for significant levels of 5% and 1%, respectively, and the values in brackets are T-values.

**Table 5 ijerph-18-08535-t005:** Global estimation results of GS3SLS.

Items	Geographical Distance		Economic–Geographical Distance	
	Dependent Variable: e-eff (1)	Dependent Variable: d-fintech (2)	Dependent Variable: e-eff (3)	Dependent Variable: d-fintech (4)
W×d-fintech	−0.605 *** (−9.02)	1.206 *** (8.51)	−0.131 *** (−3.84)	0.112 ** (2.37)
W×e-eff	1.263 *** (13.41)	−2.444 *** (−9.00)	0.691 *** (12.21)	−0.131 (−1.00)
d-fintech	0.489 *** (17.03)	-	0.345 *** (6.65)	-
e-eff	-	1.936 *** (13.82)	-	0.814 *** (6.32)
pgdp	0.055 ** (2.54)	−0.091 * (−1.94)	0.054 ** (2.24)	0.158 *** (4.10)
urb	−0.281 *** (−7.12)	0.553 *** (6.99)	−0.216 *** (−6.22)	0.214 *** (3.82)
ind	−0.405 ** (−2.85)	0.809 *** (3.04)	−0.204 (−1.64)	0.766 *** (4.05)
ope	0.237 *** (3.25)	−0.458 *** (−3.21)	0.215 *** (3.37)	−0.199 * (−1.92)
mar	−0.500 *** (−6.73)	1.010 *** (7.64)	−0.351 *** (−4.37)	1.060 *** (11.22)
den	−0.093 *** (−9.78)	0.190 *** (11.06)	−0.087 *** (−7.01)	0.200 *** (16.48)
tra	0.031 *** (2.98)	−0.060 *** (−3.02)	0.011 (1.15)	−0.033 ** (−2.30)
pos	−0.036 *** (−3.08)	0.077 *** (3.51)	−0.007 (−0.63)	0.077 *** (4.81)
N	3124	3124	3124	3124
R2	0.9493	0.9568	0.9702	0.9811
F	6122.85 (*p* = 0.000)	8106.93 (*p* = 0.000)	9820.02 (*p* = 0.000)	16,467.77 (*p* = 0.000)

Notes: *, **, *** stand for significant levels of 10%, 5% and 1% respectively, and the values in brackets are T-values.

**Table 6 ijerph-18-08535-t006:** Results of the robustness test of distance bands.

Items	W: 0–20		W: 0–40	
	Dependent Variable: e-eff (1)	Dependent Variable: d-fintech (2)	Dependent Variable: e-eff (3)	Dependent Variable: d-fintech (4)
W×d-fintech	−0.508 *** (−7.23)	0.819 *** (6.50)	−0.599 *** (−9.05)	1.345 *** (8.90)
W×e-eff	1.080 *** (11.48)	−1.597 *** (−6.21)	1.303 *** (13.57)	−2.898 *** (−10.09)
d-fintech	0.534 *** (13.58)	-	0.443 *** (19.07)	-
e-eff	-	1.608 *** (9.62)	-	2.221 *** (15.71)
pgdp	0.025 (1.12)	−0.001 (−0.03)	0.069 *** (3.16)	−0.149 *** (−2.85)
urb	−0.262 *** (−6.64)	0.430 *** (6.32)	−0.272 *** (−6.93)	0.608 *** (6.75)
ind	−0.327 ** (−2.28)	0.624 *** (2.68)	−0.370 *** (−2.63)	0.830 *** (2.74)
ope	0.244 *** (3.36)	−0.382 *** (−3.05)	0.240 *** (3.28)	−0.533 *** (−3.25)
mar	−0.545 *** (−7.00)	1.011 *** (8.87)	−0.457 *** (−6.27)	1.026 *** (6.74)
den	−0.105 *** (−10.25)	0.198 *** (12.41)	−0.085 *** (−9.12)	0.191 *** (9.97)
tra	0.025 ** (2.46)	−0.042 ** (−2.43)	0.030 *** (−9.12)	−0.067 *** (−2.92)
pos	−0.042 *** (−3.48)	0.084 *** (4.43)	−0.032 *** (−2.73)	0.073 *** (2.89)
N	3124	3124	3124	3124
R2	0.9261	0.9320	0.9523	0.9651
F	4443.21 (*p* = 0.000)	5544.52 (*p* = 0.000)	6204.77 (*p* = 0.000)	9154.76 (*p* = 0.000)

Notes: **, *** stand for significant levels of 5% and 1%, respectively, and the values in brackets are T-values.

**Table 7 ijerph-18-08535-t007:** Results of robustness test of contiguity spatial weighting matrix.

Items	W: 0–20 (Contiguity)		W: 0–30 (Contiguity)		W: 0–40 (Contiguity)	
	Dependent Variable: e-eff (1)	Dependent Variable: d-fintech (2)	Dependent Variable: e-eff (3)	Dependent Variable: d-fintech (4)	Dependent Variable: e-eff (5)	Dependent Variable: d-fintech (6)
W×d-fintech	−0.121 ** (−2.15)	0.963 *** (9.56)	−0.614 *** (−7.93)	0.869 *** (6.65)	−0.566 *** (−7.31)	1.286 *** (7.33)
W×e-eff	0.832 *** (8.70)	−0.135 (−0.67)	1.085 *** (10.54)	−1.360 *** (−6.23)	1.277 *** (11.12)	−2.897 *** (−9.51)
d-fintech	0.112 * (1.71)	-	0.679 *** (12.97)	-	0.441 *** (19.54)	-
e-eff	-	0.336 * (0.67)	-	1.258 *** (13.66)	-	2.263 *** (18.80)
pgdp	0.164 *** (6.97)	0.286 *** (7.83)	−0.011 (−0.45)	0.052 (1.56)	0.054 ** (2.34)	−0.123 ** (−2.27)
urb	−0.147 *** (−3.90)	0.061 (1.10)	−0.237 *** (−5.65)	0.312 *** (5.52)	−0.218 *** (−5.32)	0.494 *** (5.26)
ind	0.369 ** (2.37)	1.297 *** (7.08)	−0.671 *** (−4.00)	0.312 *** (4.93)	−0.403 *** (−2.62)	0.915 *** (2.70)
ope	0.298 *** (4.16)	0.198 * (1.87)	0.221 *** (2.78)	−0.266 ** (4.93)	0.227 *** (2.90)	−0.514 *** (−2.87)
mar	0.024 (0.27)	0.996 *** (11.08)	−0.823 *** (−8.85)	1.192 *** (12.06)	−0.588 *** (−7.54)	1.333 *** (7.99)
den	−0.012 (−0.94)	0.159 *** (12.94)	−0.155 *** (−11.88)	0.222 *** (18.23)	−0.112 *** (−11.44)	0.253 *** (12.46)
tra	0.008 (0.78)	−0.027 ** (−1.98)	0.029 ** (2.57)	−0.042 *** (−2.80)	0.029 *** (2.64)	−0.067 *** (−2.66)
pos	0.034 *** (0.78)	0.113 *** (7.40)	−0.045 *** (−3.37)	0.073 *** (4.44)	−0.027 ** (−2.17)	0.062 ** (2.24)
N	3124	3124	3124	3124	3124	3124
R2	0.9677	0.9876	0.9525	0.9730	0.9590	0.9718
F	8098.71 (*p* = 0.000)	22,216.75 (*p* = 0.000)	7261.85 (*p* = 0.000)	14,091.28 (*p* = 0.000)	6569.67 (*p* = 0.000)	10,025.23 (*p* = 0.000)

Notes: *, **, *** stand for significant levels of 10%, 5% and 1%, respectively, and the values in brackets are T-values.

**Table 8 ijerph-18-08535-t008:** Results of robustness test of adjusted variables.

Items	Super-SBM-GML (2003–2018)		Global-Super-SBM	
	Dependent Variable: e-eff (1)	Dependent Variable: d-fintech (2)	Dependent Variable: e-eff (3)	Dependent Variable: d-fintech (4)
W×d-fintech	−2.362 *** (−6.84)	0.887 *** (6.53)	−0.300 *** (−8.30)	1.405 *** (7.75)
W×e-eff	1.138 *** (4.60)	−0.416 *** (−3.23)	2.187 *** (2.187)	−6.721 *** (−4.89)
d-fintech	2.686 *** (11.52)	-	0.106 *** (5.62)	-
e-eff	-	0.271 *** (9.45)	-	2.242 *** (5.40)
pgdp	0.092 (0.78)	0.022 (0.46)	0.095 *** (11.91)	−0.005 (−0.08)
urb	0.291 (0.15)	0.012 (0.18)	−0.151 *** (−13.62)	0.462 *** (6.24)
ind	−0.874 (−0.96)	0.672 *** (2.93)	−0.364 *** (−8.31)	1.403 *** (6.28)
ope	2.403 *** (5.34)	−0.652 *** (−4.78)	0.111 *** (0.111)	−0.193 * (−1.84)
mar	−3.701 *** (−7.75)	1.248 *** (10.88)	0.041 (1.54)	0.626 *** (5.84)
den	−0.660 *** (−10.31)	0.213 *** (12.52)	0.001 (0.21)	0.125 *** (8.96)
tra	0.020 (0.35)	−0.032 * (−1.92)	−0.001 (−0.33)	−0.017 (−1.22)
pos	−0.293 *** (−4.30)	0.124 *** (6.92)	−0.008 ** (−2.16)	0.103 *** (6.98)
N	3124	3124	3124	3124
R2	0.8609	0.8244	0.6462	0.9000
F	1951.85 (*p* = 0.000)	2536.74 (*p* = 0.000)	608.59 (*p* = 0.000)	2795.96 (*p* = 0.000)

Notes: *, **, *** stand for significant levels of 10%, 5% and 1%, respectively, and the values in brackets are T-values.

**Table 9 ijerph-18-08535-t009:** Regional estimation results of GS3SLS (spatial weight of geographical distance).

Items	Eastern Region		Central Region		Western Region	
	Dependent Variable: e-eff (1)	Dependent Variable: d-fintech (2)	Dependent Variable: e-eff (3)	Dependent Variable: d-fintech (4)	Dependent Variable: e-eff (5)	Dependent Variable: d-fintech (6)
W×d-fintech	−0.299 *** (−5.20)	0.876 *** (6.47)	−0.822 *** (−8.17)	1.034 *** (12.31)	−0.634 *** (−6.75)	0.968 *** (8.38)
W×e-eff	1.059 *** (12.09)	−1.776 *** (−4.49)	1.191 *** (11.95)	−1.134 *** (−7.42)	1.196 *** (11.29)	−1.821 *** (−7.45)
d-fintech	0.279 *** (8.06)	-	0.728 *** (8.97)	-	0.653 *** (9.26)	-
e-eff	-	1.599 *** (5.54)	-	0.915 *** (9.47)	-	1.518 *** (9.08)
pgdp	0.011 (0.35)	0.143 ** (2.32)	0.101 ** (2.52)	−0.008 (−0.17)	−0.017 (−0.32)	0.028 (0.35)
urb	−0.136 *** (−2.83)	0.430 *** (4.61)	−0.277 *** (−4.25)	0.288 *** (3.69)	−0.049 (−0.39)	0.055 (0.30)
ind	−0.673 *** (−3.01)	1.374 *** (3.08)	−0.438 ** (−2.30)	0.757 *** (3.86)	0.122 (0.32)	−0.170 (−0.30)
ope	−0.015 (−0.10)	0.284 (1.07)	0.230 *** (2.83)	−0.167 * (−1.70)	−0.141 (−0.31)	0.225 (0.34)
mar	−0.130 * (−1.84)	0.546 *** (4.36)	−0.652 *** (−3.30)	0.886 *** (4.14)	−3.719 *** (−9.77)	5.696 *** (8.94)
den	−0.066 *** (−3.82)	0.206 *** (6.68)	−0.120 *** (−7.48)	0.143 *** (9.36)	−0.109 *** (−5.17)	0.167 *** (7.03)
tra	0.001 (0.12)	−0.043 * (−1.67)	0.053 *** (2.98)	−0.045 ** (−2.12)	0.006 (0.25)	−0.008 (−0.25)
pos	0.026 * (1.88)	0.057 * (1.84)	−0.055 *** (−3.08)	0.060 *** (2.93)	0.022 (0.60)	−0.030 (−0.54)
N	1232	1232	1221	1221	671	671
R2	0.9886	0.9800	0.9643	0.9949	0.9885	0.9959
F	10,078.46 (*p* = 0.000)	7889.74 (*p* = 0.000)	4306.01 (*p* = 0.000)	29,348.68 (*p* = 0.000)	6003.32 (*p* = 0.000)	18,775.58 (*p* = 0.000)

Notes: *, **, *** stand for significant levels of 10%, 5% and 1% respectively, and the values in brackets are T-values.

**Table 10 ijerph-18-08535-t010:** Regional estimation results of GS3SLS (spatial weight of economic–geographical distance).

Items	Eastern Region		Central Region		Western Region	
	Dependent Variable: e-eff (1)	Dependent Variable: d-fintech (2)	Dependent Variable: e-eff (3)	Dependent Variable: d-fintech (4)	Dependent Variable: e-eff (5)	Dependent Variable: d-fintech (6)
W×d-fintech	−3.384 (−1.48)	3.360 (0.81)	−23.996 *** (−7.74)	14.637 ** (2.33)	−6.770 *** (−3.06)	7.527 ** (2.28)
W×e-eff	11.573 * (1.87)	−12.526 (−1.09)	64.775 *** (8.98)	−37.579 ** (−2.39)	18.941 *** (3.44)	−20.599 ** (−2.44)
d-fintech	0.609 *** (18.63)	-	0.708 *** (18.54)	-	0.723 *** (12.40)	-
e-eff	-	1.581 *** (17.71)	-	1.102 *** (13.23)	-	1.290 *** (11.61)
pgdp	−0.119 *** (−2.93)	0.214 *** (3.56)	−0.068 * (−1.80)	0.147 ** (2.56)	−0.062 (−0.92)	0.099 (1.12)
urb	−0.122 ** (−2.04)	0.191 * (1.96)	−0.102 * (−1.77)	0.075 (0.91)	0.056 (0.41)	−0.116 (−0.65)
ind	−0.904 *** (−3.05)	1.428 *** (2.95)	−0.069 (−0.46)	0.450 * (1.84)	0.299 (0.68)	−0.368 (−0.63)
ope	−0.185 (−1.00)	0.300 (1.00)	0.297 *** (4.64)	−0.321 *** (−3.34)	−0.262 (−0.55)	0.378 (0.60)
mar	−0.405 *** (−4.46)	0.674 *** (5.03)	−0.557 *** (−3.02)	0.714 *** (3.01)	−3.975 *** (−8.67)	5.430 *** (9.60)
den	−0.169 *** (−7.75)	0.277 *** (8.21)	−0.159 *** (−9.45)	0.206 *** (10.98)	−0.126 *** (−5.17)	0.176 *** (6.39)
tra	−0.001 (−0.04)	−0.003 (−0.10)	0.028 * (1.79)	−0.033 (−1.38)	−0.020 (−0.75)	0.026 (0.70)
pos	−0.003 (−0.17)	0.013 (0.43)	−0.037 ** (−2.34)	0.053 ** (2.36)	0.043 (0.97)	−0.042 (−0.69)
N	1232	1232	1221	1221	671	671
R2	0.9296	0.9768	0.9704	0.9809	0.9179	0.9780
F	1742.55 (*p* = 0.000)	5441.93 (*p* = 0.000)	4446.35 (*p* = 0.000)	6269.19 (*p* = 0.000)	835.27 (*p* = 0.000)	3306.43 (*p* = 0.000)

Notes: *, **, *** stand for significant levels of 10%, 5% and 1%, respectively, and the values in brackets are T-values.

**Table 11 ijerph-18-08535-t011:** Period estimation results of GS3SLS (2008–2012).

Items	Geographical Distance		Economic–Geographical Distance	
	Dependent Variable: e-eff (1)	Dependent Variable: d-fintech (2)	Dependent Variable: e-eff (3)	Dependent Variable: d-fintech (4)
W×d-fintech	−0.376 *** (−6.53)	0.980 *** (14.81)	−0.118 ** (−2.20)	0.141 (1.57)
W×e-eff	1.030 *** (14.65)	−1.753 *** (−4.88)	0.260 ** (2.17)	0.132 (0.57)
d-fintech	0.370 *** (7.37)	-	0.480 *** (8.86)	-
e-eff	-	1.641 *** (4.80)	-	1.532 *** (8.82)
pgdp	0.019 (0.86)	0.021 (0.48)	0.016 (0.64)	0.023 (0.49)
urb	−0.011 (−0.28)	0.011 (0.17)	0.012 (0.29)	−0.058 (−0.86)
ind	−0.192 (−1.29)	0.782 *** (3.10)	−0.370 ** (−2.42)	0.847 *** (3.30)
ope	0.201 *** (3.96)	−0.328 *** (−2.90)	0.168 *** (3.30)	−0.271 *** (−2.75)
mar	−0.101 (−1.16)	0.349 ** (2.37)	−0.150 * (−1.65)	0.397 *** (2.58)
den	−0.055 *** (−6.19)	0.126 *** (8.46)	−0.076 *** (−7.03)	0.147 *** (9.17)
tra	−0.006 (−0.48)	0.022 (1.11)	−0.010 (−0.82)	0.018 (0.85)
pos	−0.048 *** (−3.52)	0.103 *** (4.17)	−0.065 *** (−4.44)	0.125 *** (5.00)
N	1420	1420	1420	1420
R2	0.9865	0.9940	0.9587	0.9757
F	11,161.40 (*p* = 0.000)	31,383.46 (*p* = 0.000)	3931.48 (*p* = 0.000)	7109.45 (*p* = 0.000)

Notes: *, **, *** stand for significant levels of 10%, 5% and 1%, respectively, and the values in brackets are T-values.

**Table 12 ijerph-18-08535-t012:** Period estimation results of GS3SLS (2013–2018).

Items	Geographical Distance		Economic–Geographical Distance	
	Dependent Variable: e-eff (1)	Dependent Variable: d-fintech (2)	Dependent Variable: e-eff (3)	Dependent Variable: d-fintech (4)
W×d-fintech	−0.657 *** (−5.78)	0.987 *** (5.94)	−0.108 ** (−2.52)	0.032 (0.56)
W×e-eff	1.251 *** (9.20)	−1.361 *** (−4.70)	0.607 *** (8.50)	−0.039 (−0.26)
d-fintech	0.536 *** (8.55)	-	0.522 *** (8.35)	-
e-eff	-	1.125 *** (7.52)	-	0.884 *** (6.16)
pgdp	0.099 *** (2.57)	0.009 (0.15)	0.070 * (1.85)	0.105 * (1.78)
urb	−0.520 *** (−7.88)	0.719 *** (7.44)	−0.431 *** (−7.38)	0.505 *** (5.61)
ind	−0.517 ** (−2.21)	0.982 *** (3.38)	−0.541 *** (−2.60)	1.112 *** (3.87)
ope	0.484 (1.31)	−0.401 (−0.81)	0.767 ** (2.19)	−0.629 (−1.28)
mar	−0.601 *** (−5.16)	1.009 *** (7.52)	−0.663 *** (−5.85)	1.202 *** (9.24)
den	−0.124 *** (−6.60)	0.218 *** (11.34)	−0.153 *** (−7.75)	0.257 *** (14.01)
tra	0.041 ** (2.55)	−0.052 ** (−2.42)	0.043 *** (2.81)	−0.059 *** (−2.82)
pos	−0.042 ** (−2.26)	0.101 *** (4.59)	−0.039 ** (−2.11)	0.109 *** (4.80)
N	1704	1704	1704	1704
R2	0.9424	0.9694	0.9625	0.9814
F	3052.99 (*p* = 0.000)	6857.65 (*p* = 0.000)	4760.23 (*p* = 0.000)	9732.83 (*p* = 0.000)

Notes: *, **, *** stand for significant levels of 10%, 5% and 1%, respectively, and the values in brackets are T-values.

## Data Availability

The data presented in this study are available on request from the author.

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
