# Peer review of "Spatial Interaction Spillover Effects between Digital Financial Technology and Urban Ecological Efficiency in China: An Empirical Study Based on Spatial Simultaneous Equations"

_ijerph, 2021, doi:10.3390/ijerph18168535_

Round 1

Reviewer 1 Report

First of all, I would like to thank you for the opportunity to review this paper. It is a very interesting paper and one that holds the reader's attention. The paper aims to analyze the spatial interaction spillovers between digital financial technology and urban ecological efficiency based on data from 284 Cities in China from 2008 to 2018. It is therefore an innovative and very current theme.
Overall, the paper is well written. The methodological part and the way the data were analyzed is one of the highlights.
Regarding the structure of the paper, however, I believe that a small change in item 1, dividing it into two items (Introduction and Theoretical framework) could improve the overall structure of the paper.
The question of motivation is an important point in this and all papers, but it should be included in the introduction, along with other elements, such as contextualization, for example. There is no need for an explicit contextualization item.
In the case of items 2, 3, and 4, they are well presented and very well developed.
However, the paper still presents a weakness regarding the discussion with the literature. Although there is a discussion of the results found, there is a lack of discussion about how these results relate to previous literature.
Although in Chapter 5 there is some mention of the implications of the research and the results achieved, the presence of a new discussion item before the current item 5, contemplating a closer approximation of these results with the previous literature, may bring benefits to the paper.

Author Response

We gratefully thank the editor and all reviewers for their time spend making their constructive remarks and useful suggestions, which has significantly raised the quality of the manuscript and has enable us to improve the manuscript. Each suggested revision and comment, brought forward by the reviewers was accurately incorporated and considered. Below the comments of the reviewers are response point by point and the revisions are indicated.

Point 1: Regarding the structure of the paper, however, I believe that a small change in item 1, dividing it into two items (Introduction and Theoretical framework) could improve the overall structure of the paper. The question of motivation is an important point in this and all papers, but it should be included in the introduction, along with other elements, such as contextualization, for example. There is no need for an explicit contextualization item.

Response 1: We gratefully appreciate for your valuable suggestion. In the revised draft, according to your suggestion, we divided the item 1 into two items: Introduction and Theoretical Framework. We also combine the motivation section with the literature review section of item 1 according to your suggestion.

Point 2: The paper still presents a weakness regarding the discussion with the literature. Although there is a discussion of the results found, there is a lack of discussion about how these results relate to previous literature. Although in Chapter 5 there is some mention of the implications of the research and the results achieved, the presence of a new discussion item before the current item 5, contemplating a closer approximation of these results with the previous literature, may bring benefits to the paper.

Response 2: Thank you for your rigorous consideration. We realized that there is indeed lack of discussion about how these results relate to previous literature. In the revised draft, we enriched the part about the contribution of this article. Specifically, we added the discussion of comparison with the existing literature in the “The main work and marginal contribution section” of item 2. Please see page 5, lines 192-215.

“(1) there are mutual promotion effects between digital financial technology and urban ecological efficiency, and the latter is in a relatively dominant position. The promotion of ecological efficiency by digital financial technology is in line with the previous research [9-12]. The research of this paper makes up for the blank of academic circles on the impact of ecological efficiency on digital financial technology. In other words, based on the previous research about the impact of ecological efficiency on economic development, this paper expands the impact object of ecological efficiency [22-24].(2) Both digital financial technology and urban ecological efficiency have significant spatial spillover effects; that is, digital financial technology and urban ecological efficiency of surrounding cities promote local digital financial technology and urban ecological efficiency, respectively. This is not only consistent with the research results of Shen et al. on the spatial distribution of digital financial indices [33], but also consistent with the research results of Xu et al. on the ecological efficiency spatial spillover [8]. (3) Digital financial technology of surrounding cities has an inhibitory effect on the ecological efficiency of local cities, and the improvement of the ecological efficiency of surrounding cities has a siphon effect on the local digital financial technology. Most previous studies have ignored the spatial interaction effect between the two, and the research results of this article make up for the vacancy here. (4) There is temporal and spatial heterogeneity in the intensity of the spatial interactive spillover effects between digital financial technology and urban eco-efficiency; that is, the intensity of the spatial interactive spillover effects between the two are different in the eastern, central, and western regions as well as in different periods. The heterogeneity results in this paper are similar to research of Liu et al and Shi et al. [45, 46], and the results are more comprehensive compared with other research, where they only considered one-way spatial relationship between variables.”

Reviewer 2 Report

Dear Authors,

Thank you for your interesting article. I would like to recommend you to explain more about the relationship between the financial digital technology and the ecological efficiency. Is there significant relationship between the two? 

-Could you please explain why you have used Python web crawler technology and the super-efficiency SBM-GML model ?

-Could you please position your paper within the conversation of this area of research by comparing and contrasting with similar researches?

Thank you,

M.

Author Response

We gratefully thank the editor and all reviewers for their time spend making their constructive remarks and useful suggestions, which has significantly raised the quality of the manuscript and has enable us to improve the manuscript. Each suggested revision and comment, brought forward by the reviewers was accurately incorporated and considered. Below the comments of the reviewers are response point by point and the revisions are indicated.

Point 1: I would like to recommend you to explain more about the relationship between the financial digital technology and the ecological efficiency. Is there significant relationship between the two?

Response 1: We gratefully thanks for the precious time the reviewer spent making constructive remarks. In the research hypothesis part, we explain the interaction between digital financial technology and urban ecological efficiency. Please see page 5, lines 219-231. In addition, the theoretical analysis of hypothesis 2 and hypothesis 3 also explains the relationship between the two variables.

“Digital financial technology can help improve spatial allocation and utilization efficiency of innovative resources, further improving urban ecological efficiency. The improvement of urban ecological efficiency can also promote financial innovation, absorb more financial resources, and improve digital financial technology through paths like industrial ecosystem optimization. In other words, there is an internal mechanism of mutual promotion between the two, which needs to be revealed through systematic research. Firstly, digital financial technology can optimize the momentum of urban green development and improve urban ecological efficiency through innovation, sale economy, knowledge spillover and environmental effect. Secondly, the improvement of urban ecological efficiency means that the resources invested in urban development produce more benefits than before. The improvement of urban ecological efficiency can promote the spatial aggregation of digital financial technology resources through resource aggregation, business form innovation, cost reduction and environmental optimization.”

Point 2: Could you please explain why you have used Python web crawler technology and the super-efficiency SBM-GML model?

Response 2: We gratefully appreciate for your valuable suggestion. In this revised draft, we add or enrich explanation about why we used Python web crawler technology and the super-efficiency SBM-GML model. Please see page 10, lines 403-411, and page 11, lines 441-456.

“Generally speaking, there are two main ways to obtain external data: The first is to obtain externally public data sets. For example, some scientific research institutions, enterprises, and governments will open some data. These data sets are usually relatively complete and relatively high-quality. Currently, there is no relevant data on digital financial technology at the city level in China. The second is to use crawlers tools to crawl from the Internet, such as obtaining recruitment information for a certain position from a recruitment website, renting a house website to obtain rental information in a certain area, and e-commerce website obtaining information about a certain product, etc., based on these We can do data analysis on the crawled data.”

“DEA (Data Envelopment Analysis) is the most commonly used method to measure ecological efficiency in previous studies. In this paper, urban ecological efficiency is measured using the DEA approach based on the super-efficiency SBM-GML model [48, 49]. DEA was first proposed in 1978 to evaluate the relative efficiency of a group of decision-making units with multiple inputs and outputs [50]. The distance functions of the baseline model are CCR and BCC models, but they do not consider the “Slack” phenomenon. To make up for this shortcoming, Tone put forward the SBM model and the super-efficiency SBM model in 2001 and 2002, respectively. The latter not only considers the relaxation variable but also can rank the decision-making units with the efficiency value greater than 1 [51, 52]. The Malmquist-TFP index was first introduced by Malmquist [53] and formally developed in Caves’ innovative research [54]. It is used to measure the TFP change between two periods, and the directional distance function containing undesired output is introduced into the Malmquist index to support the analysis of undesired output. In order to facilitate intertemporal comparison and overcome the problem of no viable solution, Oh included the production unit in the Global reference set and constructed the Global-Malmquist-Luenberger (GML) index [55].”

Point 3: Could you please position your paper within the conversation of this area of research by comparing and contrasting with similar researches?

Response 3: Thank you for your rigorous consideration. We realized that there is indeed lack of discussion about how these results relate to previous literature. In the revised draft, we enriched the part about the contribution of this article. Specifically, we added the discussion of comparison with the existing literature in the “The main work and marginal contribution section” of item 2. Please see page 5, lines 192-215.

“(1) there are mutual promotion effects between digital financial technology and urban ecological efficiency, and the latter is in a relatively dominant position. The promotion of ecological efficiency by digital financial technology is in line with the previous research [9-12]. The research of this paper makes up for the blank of academic circles on the impact of ecological efficiency on digital financial technology. In other words, based on the previous research about the impact of ecological efficiency on economic development, this paper expands the impact object of ecological efficiency [22-24].(2) Both digital financial technology and urban ecological efficiency have significant spatial spillover effects; that is, digital financial technology and urban ecological efficiency of surrounding cities promote local digital financial technology and urban ecological efficiency, respectively. This is not only consistent with the research results of Shen et al. on the spatial distribution of digital financial indices [33], but also consistent with the research results of Xu et al. on the ecological efficiency spatial spillover [8]. (3) Digital financial technology of surrounding cities has an inhibitory effect on the ecological efficiency of local cities, and the improvement of the ecological efficiency of surrounding cities has a siphon effect on the local digital financial.”
